# Adjuvant dendritic cell therapy in stage IIIB/C melanoma: the MIND-DC randomized phase III trial

Autologous natural dendritic cells (nDCs) treatment can induce tumor-specific immune responses and clinical responses in cancer patients. In this phase III clinical trial (NCT02993315), 148 patients with resected stage IIIB/C melanoma were randomized to adjuvant treatment with nDCs ($n = 99$) or placebo ($n = 49$). Active treatment consisted of intranodally injected autologous CD1c+ conventional and plasmacytoid DCs loaded with tumor antigens. The primary endpoint was the 2-year recurrence-free survival (RFS) rate, whereas the secondary endpoints included median RFS, 2-year and median overall survival, adverse event profile, and immunological response The 2-year RFS rate was 36.8% in the nDC treatment group and 46.9% in the control group ($p = 0.31$). Median RFS was 12.7 months vs 19.9 months, respectively (hazard ratio 1.25; 90% CI: 0.88–1.79; $p = 0.29$). Median overall survival was not reached in both treatment groups (hazard ratio 1.32; 90% CI: 0.73–2.38; $p = 0.44$). Grade 3–4 study-related adverse events occurred in 5% and 6% of patients. Functional antigen-specific T cell responses could be detected in 67.1% of patients tested in the nDC treatment group vs 3.8% of patients tested in the control group ($p < 0.001$). In conclusion, while adjuvant nDC treatment in stage IIIB/C melanoma patients generated specific immune responses and was well tolerated, no benefit in RFS was observed.

As dendritic cells (DCs) are the most potent antigen-presenting cells, presenting antigens to naive T cells, they play a pivotal role in the induction of adaptive immune responses against tumors[1]. For DC-based immunotherapy of cancer patients, autologous DCs are matured and loaded with relevant tumor antigens ex vivo and are subsequently administered to the patient to induce a tumor-specific T cell response in vivo[2]. Recent breakthroughs in immunotherapy in cancer patients mainly consist of clinical benefit from immune checkpoint inhibitors (ICI). Unfortunately, these drugs can give rise to severe immune-related toxicity due to the enhancement of non-tumor-specific immune responses against healthy cells[3]. In contrast, the toxicity profile of DC-based immunotherapy is mild[4] because T cell activation is highly antigen-specific. To date, survival benefit with DC-based immunotherapy has not been established.

Until recently, most studies with DC-based immunotherapy were performed with autologous DCs differentiated ex vivo from monocytes or CD34+ progenitors. However, the potency of these so-called monocytes-derived DCs (moDCs) may be hampered by their extensive culture period of 5–9 days with cytokines such as granulocyte-macrophage colony-stimulating factor (GM-CSF) and interleukin (IL)-4, required to differentiate the cells into DCs. Especially IL-4 potentially reduces the migration capacity of DCs[5–7]. Recently, direct isolation of the scarce naturally circulating DCs (nDCs) from blood became possible, thereby omitting the extensive culture period used for the production of moDCs[8]. After direct isolation, nDCs are manufactured by maturation and antigen loading within two days. The two major subsets of nDCs are conventional DCs (also called 'myeloid' or 'classical' DCs) and plasmacytoid DCs (pDCs). The major subset of

e-mail: Jolanda.deVries@radboudumc.nl

conventional DCs, cDC2, is characterized by CD11c and CD1c (BDCA-1) expression. pDCs express both CD303 (BDCA-2) and CD304 (BDCA-4)[9]. Manufacturing of a product of CD1c+ cDC2 and pDCs is feasible and has been tested as treatment in phase I/II clinical trials in metastatic (stage IV) melanoma and metastatic prostate cancer[10–12]. Previous studies with nDCs demonstrated that nDC treatment is safe, with little treatment-related side effects[12,13].

Of all cancer types, melanoma is by far the most studied cancer type in DC-based immunotherapy. Schadendorf et al. conducted the first, and thus far only, randomized DC-based immunotherapy trial in 2006 which did not show a survival benefit compared to dacarbazine in metastatic melanoma[14]. In the previous two decades numerous small, non-randomized, monocentric trials with DC-based immunotherapy have been performed to optimize the DC products. In 2010, the first DC-like vaccine, sipuleucel-T in men with metastatic castration-resistant prostate cancer, showed overall survival (OS) benefit and has been approved as standard treatment[15]. After decades of optimization of DC products, it is critical to re-explore DC-based immunotherapy in a randomized trial in melanoma.

Locoregional stage III melanoma is treated with surgical resection with curative intent. Unfortunately, despite complete surgical resection patients have a high risk of recurrence, resulting in 5-year OS rates between 78% and 40%[16]. Therefore, effective adjuvant therapy for this group of patients is warranted. Compared to patients with stage IV melanoma, patients with completely resected stage III melanoma harbor less tumor burden, hence show less tumor-induced immune suppression[17]. High tumor burden might hamper response to DC-based immunotherapy, as such, DC-based immunotherapy might be more successful in stage III melanoma patients. This hypothesis is supported by superior induction of antigen-specific T cells by DC-based immunotherapy in stage III compared to stage IV melanoma patients, with functional antigen-specific T cells detected after DC-based immunotherapy in 64% and 23% of patients, respectively[18,19]. In addition, a retrospective analysis of stage III melanoma patients receiving adjuvant moDC injections showed longer OS compared to their matched controls[20], which led to the initiation of this academic-initiated trial (financially supported by the Dutch National Health Care Institute). Our previous feasibility trial with combined cDC2s and pDCs in a number of stage III melanoma patients was successful in terms of production of the nDC product at the Radboud university medical center and induction of immunological response[13].

Here, we determined whether adjuvant treatment with combined cDC2 and pDC (nDC) therapy, after standard surgery in stage IIIB and IIIC melanoma patients, improves the 2-year recurrence-free survival (RFS) rate compared to placebo. At time of trial initiation, no adjuvant therapy significantly affecting survival was registered. Therefore, we included patients with completely resected stage III melanoma to receive adjuvant nDC therapy. During the enrollment phase, several drugs were approved as adjuvant therapy[21–24], causing accrual of patients in the trial to be stopped prematurely.

Here, we present the results of clinical and immunological responses, feasibility, and safety of a double-blind, randomized, placebo-controlled phase 3 trial including patients with stage IIIB and IIIC melanoma receiving adjuvant therapy with nDCs.

## Results
### Patients
Between November 2016 and November 2018, 179 patients were screened of which 148 patients were eligible. Screen failures were caused by early recurrence after surgery ($n = 23$) and lymphocytopenia ($n = 4$). The high rate of detection of early recurrences at screening, despite prior imaging, is described previously[25]. Eligible patients were randomized, 99 to the nDC treatment group and 49 to the control group (intention-to-treat population; Fig. 1). Patient and disease characteristics were equally divided over the treatment groups (Table 1). Of the 148 randomized patients, one patient withdrew consent prior to start of apheresis and was excluded from the safety analyses. Apheresis was successful in 145 of 147 patients (99%), 2 patients discontinued the study as peripheral access for apheresis could not be established despite ultrasound guidance. Four patients did not start the allocated treatment by randomization due to recurrent disease prior to the first injection. Treatment consisted of 3 biweekly intranodal injections, repeated after 6 and 12 months (*see Methods for treatment schedule*).

Of the 148 patients who were randomly assigned, 139 (94%) received at least one dose of study treatment and 124 (84%) completed the first round of treatment and the delayed-type hypersensitivity (DTH) skin test (population for immune analysis). A total of 43 patients (43.4%) in the nDC treatment group and 26 patients (53.1%) in the control group completed all three scheduled treatment cycles (Fig. 1). Of the 7 patients who withdrew consent, 5 patients withdrew consent as anti-PD-1 treatment became available as standard adjuvant treatment.

### nDC product characteristics
Patients in the nDC treatment group were treated with autologous nDCs loaded with tumor peptides and overlapping peptide pools. Manufacturing of nDC products meeting release criteria was feasible for all patients who were successfully apheresed (Supplementary Fig. 1). The average yield, purity, and viability were $59 \times 10^6$, 84%, and 93% for nDC products, respectively. In 96 (100%) patients with a successful apheresis in the nDC treatment group the number of nDCs generated was sufficient for at least 3 injections and sufficient for all 9 injections in 59 (61.5%) patients. Twenty-three patients in the nDC treatment group received at least one placebo injection due to insufficient nDC yield; 13 patients received 1 placebo injection, 1 patient received 2 placebo injections, 7 patients received 3 placebo injections, 1 patient received 5 placebo injections, and 1 patient received 6 placebo injections. All patients receiving a placebo injection in the nDC treatment group received at least 3 nDC injections, and 20 out of 23 patients received at least 6 nDC injections (range of nDC injections: 3–8).

### Clinical outcome
At data cutoff, the median duration of follow-up was 56.3 months. A total of 100 patients discontinued the trial because of disease recurrence. The primary endpoint, 2-year RFS, was not significantly different between the nDC treatment group and the control group, 36.8% (90% CI 29.6–45.7%) and 46.9% (90% CI 36.6–60.3%), respectively ($p = 0.31$, $\chi^2(1) = 1.02$). Median RFS was 12.7 months (90% CI 7.7–17.7) in the nDC treatment group vs 19.9 months (90% CI 11.7–48.5) in the control group (hazard ratio 1.25; 90% CI: 0.88–1.79; $p = 0.29$; Fig. 2). A total of 38 patients died during follow-up. The 2-year OS rate was 84.7% (90% CI 78.9–90.9%) in the nDC treatment group and 91.8% (90% CI 85.6–98.5%) in the control group ($p = 0.34$, $\chi^2(1) = 0.91$). The median OS was not reached in both treatment groups (hazard ratio 1.32; 90% CI: 0.73–2.38; $p = 0.44$).

### Induction of antigen-specific T cell responses
After the first treatment cycle, skin-test infiltrating lymphocytes (SKILs) were cultured from DTH skin test biopsies and analyzed for functionality by IFNγ production upon co-culture with autologous peripheral blood mononuclear cells (PBMCs) loaded with the relevant antigens. A DTH skin test was performed in 81 (82%) patients in the nDC treatment group and 43 (88%) patients in the control group and functional T cell analysis was performed in 73 and 26 patients, respectively (Fig. 3). In 6 of 81 (7.4%) patients in the nDC treatment group and 16 of 43 (37.2%) patients in the control group there was insufficient outgrowth of SKILs for testing. Functional antigen-specific T cell responses could be detected in 49 of 73 patients tested (67.1%) in the nDC treatment group vs 1 of 26 patients tested (3.8%) in the control

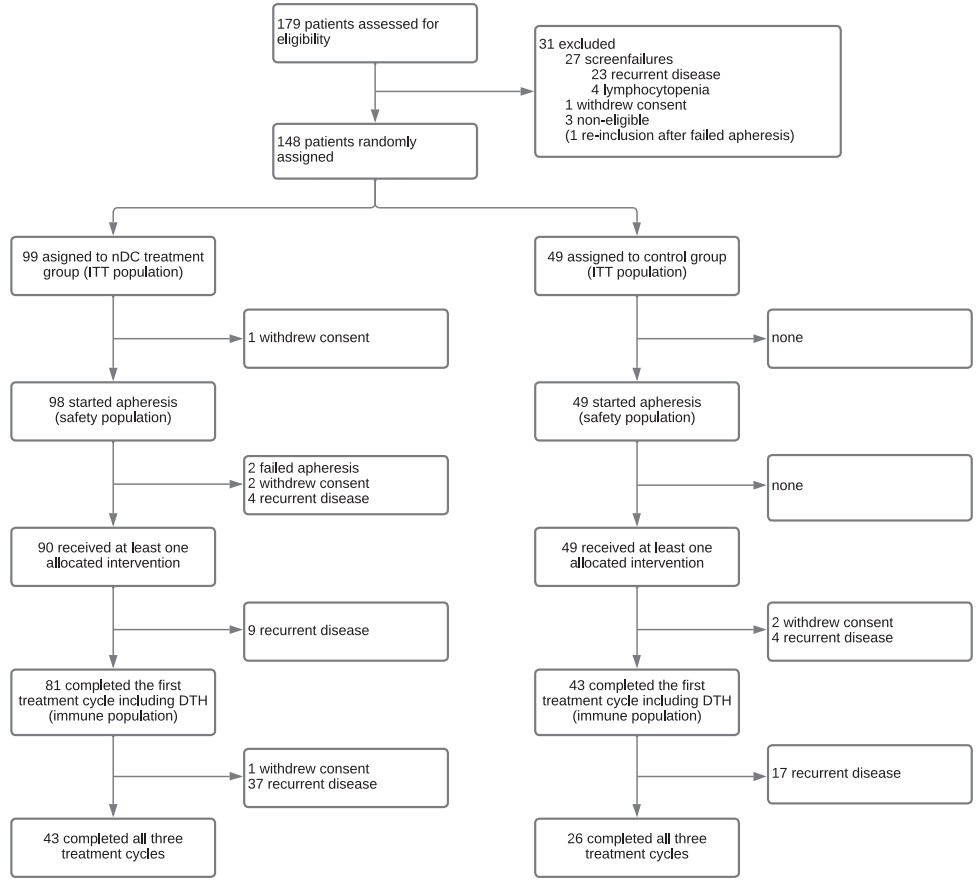

**Fig. 1 | Trial profile.** CONSORT diagram showing reasons for exclusion from the study and the number of patients included in the analyses.

group ($p = 0.0000001$, $\chi^2(1) = 28$). No clinical benefit of the presence of functional, IFNγ-producing, antigen-specific T cells was observed.

HLA-A1, HLA-A2, and HLA-B35 positive patients could be tested for antigen-specific CD8+ T cells in the SKILs and peripheral blood against 1, 5, and 2 of the used tumor antigens, respectively, by staining with dextrameric MHC complexes (Supplementary Fig. 2). Prior to start of treatment antigen-specific T cells could be detected in the peripheral blood in 7 of 56 patients tested (12.5%) in the nDC treatment group and 8 of 31 tested (25.8%) in the control group ($p = 0.20$, $\chi^2(1) = 1.6$). After the first treatment cycle, tumor antigen-specific T cells became detectable in the blood of 5 additional patients in the nDC treatment group and 3 patients in the control group.

In the SKILs, antigen-specific CD8+ T cells could be detected by dextrameric MHC complexes in 16 of 45 patients tested (35.6%) in the nDC treatment group. Of these 16 patients, SKILs of 15 patients produced IFNγ upon coculture (one not tested) and in 3 patients antigen-specific T cells could also be detected in peripheral blood. In the control arm antigen-specific T cells could be detected in 4 of 17 tested (23.5%) in the SKILs ($p = 0.55$, $\chi^2(1) = 0.36$). This included the single patient with functional reactivity in the skin test and coincided with detectable antigen-specific T cells in the peripheral blood. Individual antigen-specific T cell data is available in Supplementary Fig. 3.

**Adverse events**

Almost all patients (94% in the nDC treatment group vs 100% in the control group) experienced at least one grade 1–2 adverse event (Table 2). Apheresis-related grade 1–2 adverse events were reported in 47% in the nDC treatment group and 41% in the control group. Treatment-related grade 1–2 adverse events were reported in 79% and 82%, respectively. The most common related adverse events were flu-like symptoms, pain or hematoma at the injection site, fatigue, and

paresthesia (during apheresis). Study-related adverse events of grade 3 severity were reported by 5% of patients in the nDC treatment group vs 6% in the control group, none of the reported events occurred in more than one patient except syncope. Serious adverse events occurred in 14 patients. Only one of these serious adverse events was considered possibly related to the study in the nDC treatment group, a wound infection shortly after apheresis (before any treatment injections). In addition, one suspected unexpected serious adverse reaction (polymyalgia rheumatica grade 2) was reported as possibly related to the treatment in the nDC treatment group. No deaths were related to treatment.

## Discussion

This phase 3 trial, in which patients with resected, high-risk stage IIIB and IIIC melanoma were treated with nDCs or placebo, is the first randomized trial performed with nDC therapy. Previously, nDC therapy showed a mild toxicity profile[12,13]. Here, we could confirm that adverse events were mild and most events resolved within a few days after injection. In the nDC treatment group, 67.1% of patients mounted a functional T cell response against the antigens used for nDC loading. Unfortunately, these immunological responses did not translate into a clinical benefit. The study failed to meet the primary endpoint and the curves even show a slight, non-significant, survival disadvantage in the nDC treatment group.

The unexpected reverse survival curves might be caused by a difference in baseline gut microbiome despite stratification of patients according to the most important known prognostic disease factors and baseline characteristics. The nDC treatment group showed a relative underrepresentation of health-associated commensals, especially *Faecalibacterium prausnitzii,* which correlates with a higher risk of disease recurrence. *Faecalibacterium prausnitzii* is known to

## Table 1 | Baseline characteristics of the patients

| | DC treatment group (*n* = 99) | Control group (*n* = 49) |
|---|---|---|
| Median age, years (range) | 55 (22–80) | 56 (23–76) |
| **Sex** | | |
| Male | 56 (56.6%) | 28 (57.1%) |
| Female | 43 (43.4%) | 21 (42.9%) |
| Mean BMI (range) | 27 (18.7–38.8) | 26 (18.3–40.8) |
| **Primary tumor stage** | | |
| Unknown primary | 10 (10.1%) | 3 (6.1%) |
| T1 | 11 (11.1%) | 4 (8.2%) |
| T2 | 34 (34.3%) | 13 (26.5%) |
| T3 | 20 (20.2%) | 15 (30.6%) |
| T4 | 21 (21.2%) | 14 (28.6%) |
| Tx | 3 (3.0%) | 0 (0%) |
| **Primary tumor ulceration** | | |
| Yes | 31 (31.3%) | 19 (38.8%) |
| No | 56 (56.6%) | 26 (53.1%) |
| Unknown | 12 (12.1%) | 4 (8.2%) |
| **Nodal stage** | | |
| N1 | 44 (44.4%) | 16 (32.7%) |
| N2 | 25 (25.3%) | 16 (32.7%) |
| N3 | 30 (30.3%) | 17 (34.7%) |
| **AJCC 7th stage** | | |
| IIIB | 58 (58.6%) | 25 (51.0%) |
| IIIC | 41 (41.4%) | 24 (49.0%) |
| **Performance status** | | |
| 0 | 93 (93.9%) | 43 (87.8%) |
| 1 | 6 (6.1%) | 6 (12.2%) |
| **BRAF status** | | |
| BRAF V600 mutation | 58 (58.6%) | 27 (55.1%) |
| BRAF wildtype | 39 (39.4%) | 20 (40.8%) |
| Unknown | 2 (2.0%) | 2 (4.1%) |
| **HLA-type** | | |
| HLA-A*02 positive | 38 (38.4%) | 18 (36.7%) |
| HLA-A*02 negative | 61 (61.6%) | 31 (63.3%) |
| **Adjuvant radiotherapy** | | |
| Yes | 29 (29.3%) | 15 (30.6%) |
| No | 70 (70.7%) | 34 (69.4%) |

Data are n (%) unless otherwise stated. *AJCC* American Joint Committee on Cancer, 7th edition.

correlate with immunostimulatory effect of immune checkpoint inhibition[26,27]. Thus, the underrepresentation of this bacteria in the gut of patients in the nDC treatment group could have possibly negatively affected the outcome of this trial[28].

The standard treatment of stage III melanoma has changed during this trial, from follow-up after surgery to adjuvant treatment with anti-PD1 antibodies or BRAF/MEK inhibition. These drugs have shown a significant improvement on disease recurrence[22–24,29]. As anti-PD1 antibodies became available in the Netherlands in November 2018 as adjuvant treatment, accrual was stopped after inclusion of 148 patients as it became unethical to include patients in a placebo-controlled trial. It is highly unlikely that the main conclusion of this trial, the lack of improvement in the 2-year RFS rate with nDC therapy, was impacted by the incomplete accrual of patients as the hazard ratio is in favor of the control group.

The immunological response rate, with functional, IFNγ producing, antigen-specific T cells detected in DTH skin tests in 67.1% of the patients with nDC therapy, was in line with our previous results in stage III melanoma patients (64%) and substantially higher than in patients with stage IV disease (23%)[4,18,30]. As expected, T cell outgrowth was lower in the control group (62.8% vs 92.6%) likely due to the absence of infiltrating T cells after injection with saline as control. Insufficient outgrowth of T cells could be considered a negative result, i.e., indicate the absence of functional antigen-specific T cells in the skin biopsy. In 6 patients (7.4%) in the nDC treatment group, no outgrowth of T cells was observed. Since these 6 biopsies had to be transported for immune analyses, we speculate that transportation conditions and time hampered T cell outgrowth in these patients. Significantly more functional antigen-specific T cells were detected in the DTH skin test in the nDC treatment group, indicative that the nDCs were able to induce immune response directed against the tumor-associated antigens presented. Despite previous observations that the presence of functional specific T cells in DC-induced DTH biopsies was a valuable tool to predict clinical benefit[10,11,31], this was not confirmed in this trial.

Besides IFNγ production, we also analyzed the presence of antigen-specific CD8$^+$ T cells by staining with MHC dextramers containing tumor antigen peptides in HLA-A1, HLA-A2, or HLA-B35 positive patients. The difference in dextramer positive antigen-specific CD8$^+$ T cells found in the SKILs was less clear between the nDC treatment group and control group (35.6% versus 23.5%). In the peripheral blood at baseline, we found antigen-specific CD8$^+$ T cells in 12.5% in the nDC treatment group and 25.8% in the control group. As two times as much patients in the control group showed antigen-specific CD8$^+$ T cells at baseline this may have impacted the outcome of this trial in favor of the control arm. We have no other explanation for this difference than chance.

The importance of T cells in immunotherapy-induced tumor responses is clearly established[32]. The reasons for the absence of clinical efficacy despite the induction of functional antigen-specific T cells and difference in tumor reactivity at baseline in our trial might be the magnitude of the immunological response, the cytotoxic capacity of the induced T cells, and choice of tumor antigens. A combination of cancer testis antigens, MAGE-C2, MAGE-A3 and NY-ESO-1, and tumor-associated antigens, gp100 and tyrosinase, was chosen as they are known to be widely expressed on melanoma and are able to induce cytotoxic T cells that can lyse tumor cells. To prevent exclusion of patients based on HLA-type, peptides binding to molecules frequently observed in the Caucasian population were selected, including overlapping peptides for MAGE-A3 and NY-ESO-1, targeting both cytotoxic T cells and T helper cells. Due to the wide range of peptides and HLA-binding sites, virtually all patients should have been able to develop an immune response to an expressed tumor antigen. However, despite the induction of a functional T cell response against these cancer testis and tumor-associated antigens, the lack of survival benefit may indicate that these antigens are not the best targets. Recent advances in the field of neoantigen-approached immunotherapy, suggest that neoantigens could be more immunogenic and thereby a better target[33] (Weber, KEYNOTE-942 results presented at AACR 2023).

Another hypothesis for the lack of clinical benefit despite immunological response could be the low cytotoxic capacity of the induced T cells. Patients in this trial were treated with combined cDC2 and pDC injections, as natural DCs are considered favorable over moDCs. More recently, we developed the GMP isolation of the cDC1 subset, which might be even more potent stimulators of cytotoxic T cells. Feasibility of the clinical application of cDC1 treatment was recently shown[34]. Additionally, the cytotoxicity of induced T cells might be suppressed by IL-5 production or other immune suppressive factors, such as myeloid-derived suppressor cells and regulatory T cells, or the upregulation of inhibitory immune checkpoint molecules[35]. The addition of therapy interfering with these immunosuppressive mechanisms inhibiting the anti-tumor effect of DC-based immunotherapy is therefore of interest for future research. Combination of DC-based immunotherapy with monoclonal antibodies against the checkpoint

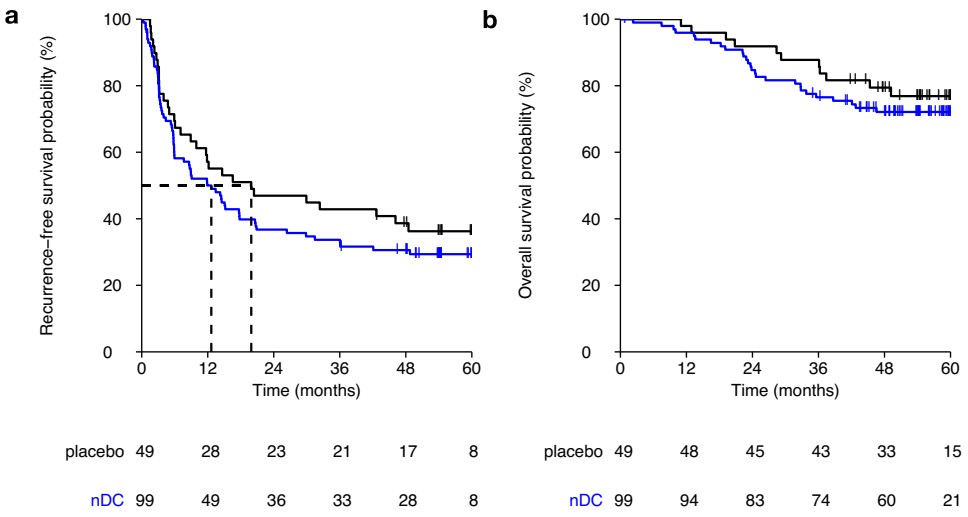

**Fig. 2 | Kaplan−Meier estimates of the survival curves of the two treatment groups. a** Recurrence-free survival and (**b**) overall survival. Source data are provided as a Source Data file.

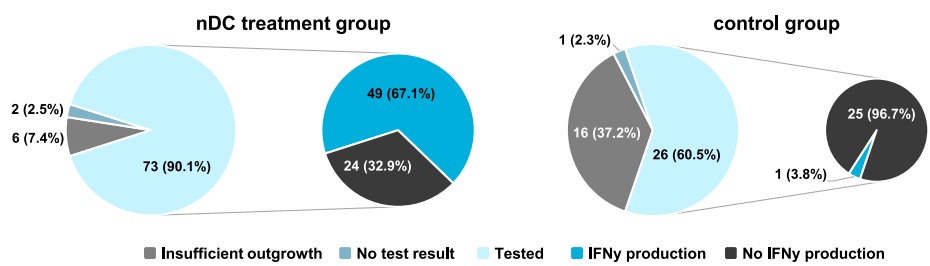

**Fig. 3 | Functional antigen-specific T cells in the immune population.** *Successful T cell outgrowth from delayed-type hypersensitivity skin test biopsies for testing is shown and whether outgrown skin-test infiltrating lymphocytes produce IFNγ upon* *co-culture with autologous PBMCs loaded with the relevant antigens for the nDC treatment group (left) and control group (right).* Source data are provided as a Source Data file.

molecule CTLA-4, which inhibits activation of T cells, already showed promising clinical responses when combined with DC-based immunotherapy[36,37]. Monoclonal antibodies interfering with the PD-1/ PD-L1 pathway inhibit T cell exhaustion in the tumor microenvironment and could therefore also enhance the anti-tumor effect of T cells induced by DC-based immunotherapy. Due to their low toxicity profile compared to anti-CTLA4 therapy, combination of anti-PD-1 treatment with DC-based immunotherapy might be worth investigating[38,39]. Further exploration of the immune responses and corresponding tumor microenvironments of patients in this trial may guide future direction of DC-based immunotherapy in cancer and other diseases.

In conclusion, execution and production of a nDC product is feasible and treatment was well tolerated. Adjuvant nDC-based immunotherapy in stage III melanoma patients induced immunological responses, but clinical benefit was not observed.

## Methods
### Study design and participants
The MIND-DC study is a double-blind, randomized, placebo-controlled phase 3 study performed in two centers in the Netherlands (Radboud university medical center, Nijmegen and Isala, Zwolle). Eligible patients were at least 18 years of age and had histologically confirmed stage IIIB or IIIC cutaneous melanoma as defined according to the American Joint Committee on Cancer 2009 classification, 7th edition[16]. Patients with an unknown primary tumor were also eligible. Patients had an

Eastern Cooperative Oncology Group (ECOG) performance status of 0 or 1, adequate hematologic, renal, and liver function, and complete surgical resection was performed no more than 12 weeks before start of study. Surgical resection consisted of a radical lymph node dissection or, after a protocol amendment (September 2017), a sentinel node procedure without an indication for radical lymph node dissection based on the outcomes of the MSLT-II trial[40]. Adjuvant radiotherapy was allowed. Patients were excluded if they had uncontrolled infectious diseases, a history of autoimmune disease (excluding skin disorders), serious conditions that may interfere with safe apheresis, or used of systemic glucocorticoids. Due to the use of keyhole limpet hemocyanin (KLH), patients with a known allergy to shellfish were excluded. The protocol was approved by the Dutch Central Committee on Research Involving Human Subjects and is in accordance with the Declaration of Helsinki as defined by the International Conference on Harmonization. Written informed consent was obtained from all patients. The first patient was enrolled on November 17, 2016, and the last patient was enrolled on November 28, 2018. The protocol is available in the Supplementary Information file.

### Randomization and masking
Patients were randomly assigned (2:1) to receive nDC therapy or placebo. Central randomization was performed using a custom R package[41] that implements the imbalance minimization technique as described by Pocock et al[42] considering disease stage (IIIB vs IIIC),

**Table 2 | Adverse events in the safety population**

| | nDC treatment group (n = 98) | | | Control group (n = 49) | | |
|---|---|---|---|---|---|---|
| | Grade 1–2 | Grade 3 | Grade 4 | Grade 1–2 | Grade 3 | Grade 4 |
| Any | 92 (94%) | 19 (19%) | 1 (1%) | 49 (100%) | 8 (16%) | 0 |
| Study-related | 81 (83%) | 5 (5%) | 0 | 42 (86%) | 3 (6%) | 0 |
| Apheresis-related | | | | | | |
| Any | 46 (47%) | 3 (3%) | 0 | 20 (41%) | 1 (2%) | 0 |
| Fatigue | 23 (23%) | 0 | 0 | 9 (18%) | 0 | 0 |
| Paresthesia | 21 (21%) | 0 | 0 | 9 (18%) | 0 | 0 |
| Dizziness | 9 (9%) | 0 | 0 | 1 (2%) | 0 | 0 |
| (Pre)syncope | 3 (3%) | 1 (1%) | 0 | 3 (6%) | 1 (2%) | 0 |
| Wound infection | 1 (1%) | 1 (1%) | 0 | 1 (2%) | 0 | 0 |
| Hypocalcemia | 0 | 1 (1%) | 0 | 0 | 0 | 0 |
| Treatment-related | | | | | | |
| Any | 77 (79%) | 3 (3%) | 0 | 40 (82%) | 2 (4%) | 0 |
| Flu like symptoms | 40 (41%) | 0 | 0 | 14 (29%) | 0 | 0 |
| Pain injection site | 33 (34%) | 0 | 0 | 17 (35%) | 0 | 0 |
| Fatigue | 32 (33%) | 0 | 0 | 20 (41%) | 0 | 0 |
| Hematoma injection site | 19 (19%) | 0 | 0 | 10 (20%) | 0 | 0 |
| Injection site reaction | 16 (16%) | 0 | 0 | 6 (12%) | 0 | 0 |
| Skin hypopigmentation | 13 (13%) | 0 | 0 | 5 (10%) | 0 | 0 |
| Headache | 8 (8%) | 0 | 0 | 2 (4%) | 0 | 0 |
| Dizziness | 6 (6%) | 0 | 0 | 5 (10%) | 0 | 0 |
| Myalgia | 5 (5%) | 0 | 0 | 1 (2%) | 0 | 0 |
| Cold | 1 (1%) | 0 | 0 | 3 (6%) | 0 | 0 |
| Hypophosphatemia | 1 (1%) | 0 | 0 | 0 | 1 (2%) | 0 |
| Syncope | 0 | 1 (1%) | 0 | 0 | 1 (2%) | 0 |
| Cataract | 0 | 1 (1%) | 0 | 0 | 0 | 0 |
| Skin infection | 0 | 1 (1%) | 0 | 0 | 0 | 0 |

Data are n (%). The safety analyses included all patients who started apheresis (including those patients in which the apheresis failed). All related adverse events that occurred in at least 5% of patients in any group and all adverse events of grade 3 or 4 severity are reported here. Patients might have had more than one event. Study-related is defined as any event related to either apheresis or treatment. Apheresis-related is defined as any event that occurs within one week of apheresis.

adjuvant radiotherapy (yes vs no), BRAF[V600] mutation status (BRAF mutant vs BRAF wildtype), and HLA-type (HLA-A02:01 negative vs HLA-A02:01 positive). Patients treatment allocation was masked for patients and clinical investigators. Only laboratory personnel, pharmacists, and statisticians were aware of group assignment.

## Outcomes
The primary endpoint was the 2-year RFS rate, defined as the percentage of patients who are alive and without recurrence of disease two years after randomization, compared to treatment with matching placebo. Secondary endpoints were median RFS, 2-year and median OS, adverse event (AE) profile, and immunological response. The additional secondary endpoints quality of life, quality adjusted life years and health economic aspects are not reported. Adverse events were recorded using the Common Toxicity Criteria for Adverse Events

version 4.03 up to 30 days after the last administration of study treatment or start of another cancer therapy, whichever occurred first. Serious adverse events believed to be related to the study treatment were still recorded after this period. Apheresis-related AEs are defined as all related AEs within one week of apheresis. AEs are considered related to the treatment/apheresis if the event was recorded as possible, probable, or definite related to the apheresis procedure by the treating physician.

## Procedures
An apheresis was performed four weeks prior to the first injection (Fig. 4). Then, patients received intranodal injections of nDCs ($3-8 \times 10^6$/injection) or placebo every 2 weeks for 3 doses, repeated after 6 and 12 months. Treatment was discontinued in case of disease recurrence, unacceptable toxicity, or withdrawal from the study. After the first 3 injections, a DTH skin test was performed. Feces collection for microbiome analyses were collected at the day of the first and the third injection and related analysis are reported in a separate manuscript[28]. Full details on the nDC product and DTH are given below. Patients were planned to be assessed for recurrence of disease every 3 months during the first 2 years and every 6 months thereafter up to 5 years. Disease assessment consisted of physical examination and CT scans. Other imaging techniques were used as clinically indicated. Recurrent disease was histologically confirmed, whenever possible.

## Dendritic cell isolation and preparation
Patients in the nDC treatment group were treated with autologous nDCs loaded with tumor peptides and overlapping peptide pools. Autologous mononuclear cells were harvested by apheresis. nDCs, consisting of pDCs and cDC2s, were isolated with the fully automated and closed immunomagnetic CliniMACS Prodigy® isolation system (Miltenyi Biotec, Bergisch Gladbach, Germany) with magnetic bead-coupled antibodies (Miltenyi Biotec) following the manufacturer's guidelines. cDC2s were labeled with CD1c-biotin and B cells and monocytes were depleted using magnetic bead-coupled CD19 and CD14 antibodies, respectively. Subsequently, cDC2s were positively selected with magnetic bead-coupled anti-biotin antibodies and pDCs were positively selected with magnetic bead-coupled CD304 antibodies. This procedure resulted in purified nDCs.

nDCs were cultured overnight in 6 wells plates at 37° 5% $CO_2$ at a concentration of 1.5 x $10^6$ cells/ml with 800IU/ml recombinant human GM-CSF and 10 ng/ml recombinant human IL-3 in TexMACS GMP medium (all Miltenyi Biotec) supplemented with 2% human serum (Sanquin, Amsterdam, the Netherlands), 10 µg/ml KLH (Immucothel, Biosyn Arzneimittel GmbH, Fellbach, Germany) for immunomonitoring and MACS® GMP-grade PepTivators®, overlapping peptide pools of MAGE-A3 and NY-ESO-1 (Miltenyi Biotec), consisting of 15-mer peptides with 11 amino acids overlap, covering the sequence of the entire antigen. After overnight culturing, nDCs were activated for 6 h with 15 µl/ml protamine HCl (Meda Pharma, Amstelveen, the Netherlands)/mRNA (Universitätsklinikum Erlangen, Erlangen, Germany) (protamine/mRNA). Protamine/mRNA was generated by premixing 10 µg protamine HCl (5000 IU/ml = 50 mg/ml) with 5 µg mRNA for 10 minutes. After 3 h of maturation, viability and phenotyping were assessed and a mix of fourteen peptides, consisting of gp100, tyrosinase, MAGE-C2, MAGE-A3, and NY-ESO-1 (all Leiden University Medical Center, Leiden, the Netherlands or Miltenyi Biotec), were added for the last 3 h of maturation (Supplementary Table 1). After peptide loading, cells were washed twice with injection liquid (NaCl 0.9% supplemented with 5% Albuman (final concentration)) to remove excess peptides and were cryopreserved in TexMACS medium containing 10% dimethyl sulfoxide (DMSO; WAK-chemie Medical GmbH, Steinbach, Germany) and 40% Albuman (Sanquin), stored below −150 °C, and thawed on the day of administration. The procedure had to give rise to mature nDCs

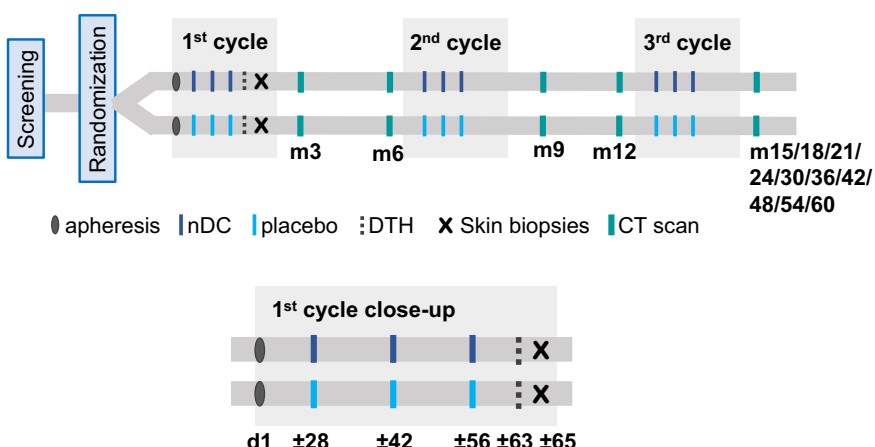

**Fig. 4 | Treatment schedule.** Randomized patients underwent an apheresis to harvest natural dendritic cells (nDCs). A treatment cycle consisted of 3 nDC injections or placebo injections. The first cycle was followed by a delayed-type hypersensitivity (DTH) skin test including skin biopsies 2 days later. Patients were assessed for recurrence of disease with CT scans.

meeting the following release criteria: sterile (tested by Eurofins Bactimm, Nijmegen, the Netherlands), free of endotoxins, >50% viability, >50% CD83 expression and a potency index >2. The potency index was defined as T cell activation in a mixed lymphocyte reaction of peripheral blood lymphocytes (PBLs) with mature nDCs/T cell activation of control PBLs without nDCs. One nDC product consisted of at least $3 \times 10^6$ nDCs and up to a maximum of $8 \times 10^6$ nDCs per dose. From the apheresis starting material, a batch of nine nDC products and 1 DTH product were manufactured. If an insufficient number of cells were obtained from the isolation and culturing procedure to generate nine nDC products, for all three treatment cycles, patients received placebo injections during the last injections. Placebo injections consisted of NaCl 0.9% supplemented with 25% Albuman (Sanquin). An experienced radiologist administered the nDC product or placebo intranodally in a radiologically tumor-free lymph node under ultrasound guidance.

## Purity and phenotype assessment

Purity and phenotype of nDCs after immunomagnetic isolation were determined by flow cytometry with a FACSVerse® (BD biosciences, San Jose, CA) or MACS Quant® (Miltenyi Biotec). For this purpose, the following primary monoclonal antibodies and the appropriate isotype or fluorescence minus one controls were used: anti-CD1c-Viobright-FITC, anti-BDCA2-PE, anti-CD123-APC, anti-CD20-PE-Vio770, anti-CD45-APC-Vio770, anti-CD14-Viogreen, anti-FcεRI-BioBlue, anti-CD14-FITC, anti-CD15-PE, anti-CD56-APC, anti-CD3-VioBlue, anti-HLA-ABC-APC, anti-HLA-DR/DP/DQ-APC, anti-CCR7-APC, anti-CD80-APC, anti-CD83-APC, and anti-CD86-APC (all Miltenyi Biotec). Details are depicted in Supplementary Table 2. The purity of the nDC product was defined as the percentage of nDCs (sum of $CD123^+BDCA2^+$ pDC plus $CD1c^+CD20^-$ cDC2) of all viable cells in the nDC product. After 6 h of protamine/mRNA stimulation, cytokine production of nDCs was measured in the supernatant by cytometric bead array according to the manufacturer's instruction (Miltenyi Biotec).

## Tumor antigen-specific and functional T cells

DTH skin tests were performed 1–2 weeks after the third injection, as described previously[43]. Depending on randomization, nDCs ($0.1–1.0 \times 10^6$ cells) or placebo was injected intradermally at the back of the patient at 4 sites. Two days after injection, skin punch biopsies (6 mm) were obtained from all injection sites to assess T cell responses. The biopsy specimens were cut in half; one half was cryopreserved, and the other half was cut and cultured for 2–5 weeks in IL-2 (Novartis, Basel, Switzerland) containing RPMI medium. After culturing, the

SKILs were tested for the presence of T cells specific for the different tumor antigens used. To test T cell functionality, SKILs were cocultured with autologous PBMCs pulsed with the different relevant peptides, overlapping peptide pools, carcinoembryonic antigen (negative control), or no peptide (negative control). Production of IFNγ was measured in the supernatants by cytometric bead array according to the manufacturer's instruction (eBioscience, Vienna, Austria, or BD Biosciences) after 24 h of coculture. Functional antigen-specific T cells were considered present if IFNγ production after co-culturing with antigen-loaded autologous PBMCs was at least 50 pg/ml higher with tumor antigens than an irrelevant control antigen.

SKILs and PBMCs of patients with HLA-types HLA-A1, HLA-A2 and HLA-B35 were stained with dextrameric-MHC complexes containing MAGE-A3 peptide in HLA-A1 positive patients; gp100, tyrosinase, NY-ESO-1, MAGE-C2, and MAGE-A3 peptide in HLA-A2 positive patients; and NY-ESO-1 and MAGE-A3 peptide in HLA-B35 positive patients. Dextrameric-MHC complexes containing irrelevant peptide were used as correction for background binding.

## Statistical analysis

To detect an improvement in the 2-year RFS rate from an estimated 50% to 70%, with a power of 80% and two-sided α level of 0.05, we planned to randomly assign 210 patients. The study was stopped prematurely after inclusion of 148 patients as adjuvant treatment with anti-PD1 antibodies became available as standard of care in the Netherlands in November 2018. RFS and OS was censored at the last time of follow-up except for patients who switched to anti-PD1 treatment, who were censored at the day the decision was made to stop study treatment.

The Kaplan-Meier method was used to estimate median RFS and OS distributions and the 90% CI of these estimates. A comparison between the groups was made using the log-rank test. Hazard ratios were estimated with a Cox proportional hazards model, stratified by stage of the disease, adjuvant radiotherapy, BRAF mutation status, and HLA-type. RFS was defined as the time between randomization and the date of first recurrence (local, regional, or distant metastasis) or death, whichever occurred first. P values for differences between fractions (such as the fraction of patients showing an immune response in the nDC treatment group versus the control group) were calculated by means of the chi-square test with Yates' continuity correction. When events had not occurred, survival was censored at the date of last follow-up. We calculated median follow-up using the inverse Kaplan–Meier method. Efficacy analysis was performed on the intention-to-treat population, defined as all eligible patients assessed in the group

they were allocated by randomization. The safety population consisted of all patients who at least started apheresis. All statistical analysis was performed using the R platform for statistical computing, version 4.2. Survival data were analyzed using the R package "survival" which is included in the R platform, and visualized using the R packages "survminer", version 0.4.9., and "ggplot2", version 3.1.1. *P* values of less than 0.05 were considered significant. This trial is registered with EudraCT, number 2015-005322-19 (preregistered January 18, 2016), and ClinicalTrials.gov, number NCT02993315.

## Reporting summary

Further information on research design is available in the Nature Portfolio Reporting Summary linked to this article.

## Data availability

Source data for all figures (i.e., the numbers being tabulated or graphically displayed) are provided with this paper. The study protocol is available in the Supplementary Information file. Individual participant data are not made publicly available owing to privacy and ethical restrictions. Specific requests for access to raw and/or analyzed data should be sent to the corresponding author. Data requests will be reviewed within 3 months by the principal investigators of the trial on the basis of scientific merit, ethical review, available resources and regulatory requirements. Any data and materials that can be shared will require approval from the Institutional Review Board and a data or material transfer agreement. If all agreements are in place, anonymized data will be made available for reuse for a prespecified time. The remaining data are available within the Article, Supplementary Information or Source Data file. Source data are provided with this paper.

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

## Acknowledgements

The authors thank the patients for their participation in the clinical trial and they thank Kiek N. Verrijp for experimental support and Eline F. Smits and Janita Post – Bakker for data management. The authors acknowl-edge the FLIMS faculty of the Workshop on "Methods in Clinical Cancer Research" 2014 for assistance on writing the protocol. The study was designed by the principal investigators at the Radboudumc. The study was funded by the Dutch National Health Care Institute (Zorginstituut Nederland), ZonMW, and Stichting Afweer Tegen Kanker (ATK), and in-kind support by Miltenyi Biotec. The funders of the study had no role in study design, data collection, data analysis, data interpretation, or writing of the report. The corresponding author had full access to all the data in the study and had final responsibility for the decision to submit for publication.

## Author contributions

Conception and design: KFB, GS, WRG, JHWdW, TJMdW, CGF, IJMdV. Laboratory analyses: GS, ALdG, AJdB, KJHB, TDdB, MAMON, TGMvO, CJP, JMP, NMS, MWMMvdR, VdR, CA, MBr, KP, AD, IJMdV. Clinical involvement: KFB, MBl, WWvW, SHdB, WSvM, MMvR, SJC, BJK, JHAC, IMNW, AAvdV, DJvG, JEMW, JJB, JBAGH, MJBS, RHTK, MFB, EHJA, MG, JN, JHWdW, JWBdG, WRG. Data collection: KFB, GS, MBl, WWvW, SHdB. Data analysis and interpretation: KFB, GS, JT, IJMdV. Preparation and critical revision of the paper: KFB, GS, JT, IJMdV. Final approval of paper: all authors.

## Competing interests

K.F.B.: consultancy fees (all paid to institute) from MSD and Pierre Fabre. A.A.v.d.V.: consultancy fees (all paid to institute) from BMS, MSD, Pierre Fabre, Merck, Pfizer, Novartis, Sanofi, Ipsen, Eisai, Roche. CA/MBr/KP/AD: employees of Miltenyi Biotec. J.W.B.d.G.: received personal fees outside the submitted work from Bristol-Myers Squibb, Roche, Pierre Fabre, Servier, MSD, and Novartis. J.B.A.G.H.: advisory relationships with Amgen, AstraZeneca, Bayer, Bristol-Myers Squibb, Celsius Therapeutics, GSK, Immunocore, Ipsen, MSD, Merck Serono, Novartis, Neon Ther-apeutics, Pfizer, Roche/Genentech, Sanofi, and Seattle Genetics and has received research grants not related to this paper from Novartis, BMS, MSD, and Neon Therapeutics. W.R.G.: speaker fees from MSD; advisory role (institutional) for Bristol-Myers Squibb and Bayer; research grants (institutional) from Astellas, Bayer, Janssen-Cilag, MSD. The remaining authors declare no competing interests.

## Additional information

**Peer review information** *Nature Communications* thanks Jessica Flynn, Craig Slingluff and the other, anonymous, reviewer(s) for their con-tribution to the peer review of this work. A peer review file is available.

**Kalijn F. Bol**[1,2], **Gerty Schreibelt** [1,15], **Martine Bloemendal**[1,2,15], **Wouter W. van Willigen**[1,2,15], **Simone Hins-de Bree**[1], **Anna L. de Goede**[3], **Annemiek J. de Boer**[1], **Kevin J. H. Bos**[1], **Tjitske Duiveman-de Boer**[1], **Michel A. M. Olde Nordkamp**[1], **Tom G. M. van Oorschot**[1], **Carlijn J. Popelier**[1], **Jeanne M. Pots**[1], **Nicole M. Scharenborg**[1], **Mandy W. M. M. van de Rakt**[1], **Valeska de Ruiter**[1], **Wilmy S. van Meeteren**[4], **Michelle M. van Rossum**[4], **Sandra J. Croockewit**[5], **Bouke J. Koeneman** [1], **Jeroen H. A. Creemers** [1], **Inge M. N. Wortel** [1,6], **Caroline Angerer**[7], **Mareke Brüning**[7], **Katja Petry**[7], **Andrzej Dzionek**[7], **Astrid A. van der Veldt**[8], **Dirk J. van Grünhagen** [9], **Johanna E. M. Werner**[10], **Johannes J. Bonenkamp**[10],

John B. A. G. Haanen ⬤ [11], Marye J. Boers-Sonderen[2], Rutger H. T. Koornstra[2], Martijn F. Boomsma[12], Erik H. J. Aarntzen[13], Martin Gotthardt[13], James Nagarajah[13], Theo J. M. de Witte ⬤ [1], Carl G. Figdor ⬤ [1], Johannes H. W. de Wilt ⬤ [10], Johannes Textor ⬤ [1,6,16], Jan Willem B. de Groot[14,16], Winald R. Gerritsen ⬤ [2,16] & I. Jolanda M. de Vries ⬤ [1] ✉

[1]Medical Biosciences, Radboud Institute for Medical Innovation, Radboud university medical center, Nijmegen, The Netherlands. [2]Department of Medical Oncology, Radboud university medical center, Nijmegen, The Netherlands. [3]Department of Pharmacy, Radboud university medical center, Nijmegen, The Netherlands. [4]Department of Dermatology, Radboud university medical center, Nijmegen, The Netherlands. [5]Department of Hematology, Radboud university medical center, Nijmegen, The Netherlands. [6]Department of Data Science, Institute for Computing and Information Sciences, Radboud University, Nijmegen, The Netherlands. [7]Miltenyi Biotec, Bergisch Gladbach, Germany. [8]Departments of Medical Oncology and Radiology & Nuclear Medicine, Erasmus Medical Center Cancer Institute, Rotterdam, The Netherlands. [9]Department Surgical Oncology, Erasmus Medical Center Cancer Institute, Rotterdam, The Netherlands. [10]Department Surgical Oncology, Radboud university medical center, Nijmegen, The Netherlands. [11]Department of Medical Oncology, The Netherlands Cancer Institute, Amsterdam, The Netherlands. [12]Department of Radiology, Isala Oncology Center, Zwolle, The Netherlands. [13]Department of Medical Imaging, Radboud university medical center, Nijmegen, The Netherlands. [14]Department of Medical Oncology, Isala Oncology Center, Zwolle, The Netherlands. [15]These authors contributed equally: Gerty Schreibelt, Martine Bloemendal, Wouter W. van Willigen. [16]These authors jointly supervised this work; Johannes Textor, Jan Willem B. de Groot, Winald R. Gerritsen ✉e-mail: Jolanda.deVries@radboudumc.nl

