## [Peer Review File · Nature Communications]

Adjuvant dendritic cell therapy in stage IIIB/C melanoma: the MIND-DC randomized phase III trialREVIEWER COMMENTS

Reviewer #1 (Remarks to the Author): with expertise in clinical trial study design, biostatistics

Here, De Vries and colleagues present a randomized phase III trial of 148 patients assessing adjuvant dendritic cell therapy in stage IIIB/V melanoma. The potential of natural dendritic cells (nDCs) treatment to improve relapse free survival and tumor-specific immune responses compared to placebo was evaluated. There did not prove to be a clinical benefit to nDCs, however, it was shown that the nDC group had more functional antigen-specific T cells and that this treatment was well tolerated in the patient population. Although the trial was stopped early due to the approval of several adjuvant therapies, valuable information regarding the role of nDCs was discovered. There are several items that should be addressed to improve the quality and rigor of this scientific report. Some questions and suggestions are detailed below:

Comments:

- 1) In the introduction it is mentioned that nDCs have been successful in treating metastatic castrate resistant prostate cancer (mCRPC), however, in the discussion it says that the nDCs may have been unsuccessful in this trial due to the choice of tumor antigens. How were antigens/targets chosen to make nDCs successful for mCRPC? Comment on and compare the success of nDCs for this population with the stage IIIB/C melanoma population in the discussion.
- 2) It was mentioned that some patients in the nDC treatment arm had to have several placebo injections. Could this have made nDC less successful as a treatment option? Please elaborate on the implications (if any) that this may have.
- 3) Can p-values be added to Table 1 to show there is no difference between cohorts? Or was this not done because it was not specified in the protocol?
- 4) In the "Clinical outcome" section the numbers listed of those who discontinued the trial because of disease recurrence does not match the consort diagram (Figure 1).
- 5) Remove percentages next to reporting the total who died. Using raw percentages ignores censoring and is misleading
- 6) In the "Adverse events" section it lists that 5% and 6% of study-related grade 3 adverse

events were reported in the nDC and control groups, respectively. Where are these percentages shown in Table 2? They do not align

7) In the Discussion it states that “The main conclusion of this trial, the lack of improvement in the 2-year RFS rate with nDC therapy, was not impacted by the incomplete accrual of patients”. How can the authors be sure of this when the accrual was stopped? This lower sample size was not powered to detect this conclusion. This should be reworded.

8) A limitation should be added regarding that the sample size powered in the protocol was not achieved, thus this may affect the conclusions of the trial.

9) Why were patients with an unknown primary tumor also eligible for inclusion? Does this make the population heterogenous? Please elaborate in the “Study design and participants” section.

10) The protocol specifies reporting a 90% confidence interval, but the manuscript “Statistical analysis” section reports using 95% confidence intervals. This should be harmonized.

11) What is meant by “P values of point estimates” in the “Statistical analysis” section? Please clarify and elaborate.

12) Survival was censored at last follow up or time of switch to adjuvant therapy for those in the control arm that switched. Please rectify this in the “Statistical analysis” section.

Reviewer #2 (Remarks to the Author): with expertise in melanoma, immunotherapy, clinical trials

The investigators have performed a phase III placebo-controlled clinical trial in 148 patients with resected stage IIIB/C melanoma, with 2:1 randomization to treatment with natural DCs vs. placebo. Patients were stratified by stage, radiotherapy, BRAF status, and HLA-A0201 status. The nDCs were loaded with tumor antigens and injected intranodally. Adverse events were similar for treatment and placebo. T cell responses were detected in most patients, but there was no significant difference in recurrence-free survival. This is a new approach using natural circulating DCs (nDCs) rather than monocyte-derived DCs generated in vitro. They show that the DCs primarily include conventional DC2s (cDC2s) and plasmacytoid DCs (PDCs). A prior feasibility study by this group supported the study design. Patients with AJCC v7 stage IIIB-C melanoma were enrolled. The primary endpoint was 2 year RFS. Enrollment

was discontinued early because of new approved therapies. This paper reports the final results with of this trial. The nDC therapy was feasible, with only about 4% not treated due to unsuccessful apheresis or recurrence before the first injection. The groups were well-matched. AEs were similar in the DC and placebo groups, supporting safety of this vaccine strategy.

The nDC were loaded with overlapping peptide pools of MAGE-A3 and NY-ESO-1 during overnight culture with GM-CSF, IL-3 and KLH, then with 14 peptides from melanocytic differentiation antigens (gp100 and tyrosinase) and cancer testis antigens (MAGE-C2, MAGE-A3, NY-ESO-1), including 11 short peptides restricted by Class I MHC (6 HLA-A2.1), and 3 longer peptides presented by class II MHC.

The RFS was no different between nDC and placebo, with a weak trend to shorter RFS with the DC vaccine. Similarly, OS was not different, with an even weaker trend favoring placebo. T cell responses to vaccine antigens were assessed by measuring IFN γ production of lymphocytes cultures from a skin test after the first cycle. Of those evaluable by this method, there was a dramatically higher T cell response rate in the DC vaccine group. Overall, 67% of the 90% evaluable by this method had T cell responses in this assay.

The authors are to be congratulated on performing what appears to be the first randomized multicenter placebo controlled trial of nDC + peptides vs placebo. The study was closed with about 70% of target enrollment, but the lack of clinical benefit for the nDC arm is not likely to be have been altered by completing the target enrollment. The presented data certainly support immunogenicity of the nDC vaccine.

There are some aspects of the study that might be enhanced to understand the outcomes more clearly.

1) The immune analysis is based on in vitro culture of T cells accumulating at a site of skin test (DTH) and stimulated with antigen-pulsed PBMC. While these data support immunogenicity in the nDC patients, compared to the placebo patients, it would be helpful to use a measure of immunogenicity that may be compared to prior experience with other vaccine strategies, such as analysis ex vivo of circulating T cells. Was blood collected that could be assayed by ELIspot or other functional assays?

2) Also, the T cell responses are not detailed in terms of antigen reactivity or CD8 vs CD4 T

cell reactivities. Since most patients were HLA-A2.1 negative, and since 15mer peptides may not induce CD8 T cells as readily as CD4 T cells, it would be helpful to understand the immune reactivity as a function of HLA expression.

3) The nDC used in this approach were primarily cDC2 and PDC. These are certainly good to include, but cDC1 are commonly considered best for induction of CD8 T cells either directly or after cross-presentation. What were the reasons for not including cDC1s?

4) Since the presumption of the authors that nDC are preferable to use, compared to monocyte-derived DC, do the authors have data on markers of DC activation and maturation, and cytokine production compared to that achieved with monocyte-derived DC? Also the potency of the mature DC appears low in panel D of Supplemental Figure 3, panel D: is there a comparator group?

5) The discussion addresses some relevant considerations, but it would be valuable to elaborate on whether lack of clinical benefit may relate to the magnitude or function of the induced T cells. The companion paper about microbiome is interesting, but it is difficult to understand why there would be a significant mismatch in microbiome in a multicenter trial.

Also, a few minor comments:

a) The Kaplan-Meier curve in Figure 2A is labelled Progression-free survival, but probably should be corrected to "recurrence-free survival".

b) Please clarify details about the overlapping peptide pools for NY-ESO-1 and MAGE-A3? Were these 15 mers overlapping by 11?

c) The data in S Fig 3, Panel A are not clear: It is not clear what the Y axis represents in panel A.

d) It would be helpful to add a protocol schema with a timeline of interventions, to follow what was done in the study.

Reviewer #3 (Remarks to the Author): with expertise in melanoma, immunotherapy, clinical trials

This is an unsuccessful phase III clinical trial of a DC product. The authors claim that this is being presented such that a companion paper describing the translational science of why the trial was unsuccessful is placed into the proper context.

1. Why was apheresis unsuccessful in two patients?
2. Table 1: Unfortunately, the investigators still used AJCC version 7. Lymph node disease suggests better patient characteristics in the treated arm. It would be worthwhile to comment on outcome based on lymph node status in the discussion.
3. It is noted that the study failed to meet the endpoint and there may be a slight disadvantage in the DC-treated group. There was no clinical benefit of IFN γ producing antigen specific T cells. The authors speculated that this is secondary to gut microbiome, but there should be more of a link to the companion paper. The clinical trial paper should be concise as a stand-alone paper.
4. What was the grade 4 reaction in the treatment group?
5. The authors noted: "Significantly more functional antigen-specific T cells were detected in the nDC treatment group, indicative that the nDCs were able to induce immune response directed against the tumor-associated antigens presented". This should be tested via an assay.
6. Supplementary figure 3 should also be divided into the two groups
7. Minor comment: Personal preference would be that supplementary Figure 1 is in the manuscript. The description of the trial is essential to have readily available to the reader of the paper. It could be combined with the current Figure 1.

REVIEWER COMMENTS MIND-DC

Reviewer #1 (Remarks to the Author): with expertise in clinical trial study design, biostatistics

Here, De Vries and colleagues present a randomized phase III trial of 148 patients assessing adjuvant dendritic cell therapy in stage IIIB/V melanoma. The potential of natural dendritic cells (nDCs) treatment to improve relapse free survival and tumor-specific immune responses compared to placebo was evaluated. There did not prove to be a clinical benefit to nDCs, however, it was shown that the nDC group had more functional antigen-specific T cells and that this treatment was well tolerated in the patient population. Although the trial was stopped early due to the approval of several adjuvant therapies, valuable information regarding the role of nDCs was discovered. There are several items that should be addressed to improve the quality and rigor of this scientific report. Some questions and suggestions are detailed below:

Comments:

1) In the introduction it is mentioned that nDCs have been successful in treating metastatic castrate resistant prostate cancer (mCRPC), however, in the discussion it says that the nDCs may have been unsuccessful in this trial due to the choice of tumor antigens. How were antigens/targets chosen to make nDCs successful for mCRPC? Comment on and compare the success of nDCs for this population with the stage IIIB/C melanoma population in the discussion.

In our opinion, we stated in the introduction that the feasibility trials in metastatic (stage IV) melanoma and metastatic prostate cancer were successful in terms of production of the nDC product at the Radboud university medical center and induction of immunological responses. We explicitly did not state that a clinical benefit was observed since these trials were small, not randomized, and were lacking a control arm. The observed immunological responses (patients with specific functional T cells in DTH) in these feasibility trials is similar to the here observed functional immunological responses. In this randomized trial, these responses do not correlate with clinical outcome as observed in the feasibility studies with comparable antigens. In the feasibility trials as well as the current randomized trial in melanoma, non-neoantigens were used for antigen loading of the nDC. Despite the induction of functional T cell responses against the cancer testis and tumor-associated antigens, the lack of survival benefit may indicate that these antigens are not the best targets. Recent advances in the field of neoantigen-approached immunotherapy, suggest that neoantigens could be more immunogenic and thereby a better target. This argument is present in the discussion.

2) It was mentioned that some patients in the nDC treatment arm had to have several placebo injections. Could this have made nDC less successful as a treatment option? Please elaborate on the implications (if any) that this may have.

This is a valid question from the reviewer.

The treatment schedule, of 3 times 3 nDC injections, was chosen decades ago with the idea that 'booster injections' might be needed after a while. In previous studies we performed DTH skin test after each cycle and could detect additional immunological responses after the 2nd or 3rd cycle, indicative for the effectiveness of 'booster injections'. As the risk of disease recurrence in stage III melanoma patients is the highest the first one to two years after resection, we decided to keep the 3 times 3 treatment schedule.

Of the 23 patients who received at least one placebo injection, 20 received 6, 7, or 8 nDC injections with 3, 2, or 1 placebo injection(s), respectively. This is now stated more clearly in the manuscript. Of the 3 patients who received less than 6 nDC injections, one patient received 5 nDC injections + 1 placebo, one patient received 3 nDC injections and 6 placebo injections, and one patient received 4 nDC injections + 5 placebo injections.

Overall, only 3 patients received less than 6 nDC injections due to low nDC yield from a single apheresis, we consider it highly unlikely this would have impacted the efficacy of nDC treatment in the entire study population.

3) Can p-values be added to Table 1 to show there is no difference between cohorts? Or was this not done because it was not specified in the protocol?

We have calculated p-values for Table 1 as requested by the reviewer; all p-values were above 0.1 and so did not indicate any significant imbalances for the covariates. However, we would prefer not to include these p-values in the manuscript. Although we are aware that this is common practice, it is also widely understood to be a misleading use of inferential quantities for descriptive purposes, and is discouraged for example by the CONSORT guidelines.

Table 1. Baseline characteristics of the patients.

	DC treatment group (n=99)	Control group (n=49)	
Median age, years (range)	55 (22-80)	56 (23-76)	p=0.6
Sex			p>0.9
Male	56 (56.6%)	28 (57.1%)	
Female	43 (43.4%)	21 (42.9%)	
Mean BMI (range)	27 (18.7-38.8)	26 (18.3-40.8)	p=0.5
Primary tumor stage			p=0.5
Unknown primary	10 (10.1%)	3 (6.1%)	
T1	11 (11.1%)	4 (8.2%)	
T2	34 (34.3%)	13 (26.5%)	
T3	20 (20.2%)	15 (30.6%)	
T4	21 (21.2%)	14 (28.6%)	
Tx	3 (3.0%)	0 (0%)	
Primary tumor ulceration			p=0.6
Yes	31 (31.3%)	19 (38.8%)	
No	56 (56.6%)	26 (53.1%)	
Unknown	12 (12.1%)	4 (8.2%)	
Nodal stage			p=0.4
N1	44 (44.4%)	16 (32.7%)	
N2	25 (25.3%)	16 (32.7%)	
N3	30 (30.3%)	17 (34.7%)	
AJCC 7th stage			p=0.4
IIIB	58 (58.6%)	25 (51.0%)	
IIIC	41 (41.4%)	24 (49.0%)	
Performance status			p=0.2
0	93 (93.9%)	43 (87.8%)	
1	6 (6.1%)	6 (12.2%)	
BRAF status			p=0.7
BRAF V600 mutation	58 (58.6%)	27 (55.1%)	
BRAF wildtype	39 (39.4%)	20 (40.8%)	
Unknown	2 (2.0%)	2 (4.1%)	
HLA-type			p=0.8
HLA-A*02 positive	38 (38.4%)	18 (36.7%)	
HLA-A*02 negative	61 (61.6%)	31 (63.3%)	
Adjuvant radiotherapy			p=0.9
Yes	29 (29.3%)	15 (30.6%)	

No	70 (70.7%)	34 (69.4%)	
----	------------	------------	--

4) In the "Clinical outcome" section the numbers listed of those who discontinued the trial because of disease recurrence does not match the consort diagram (Figure 1).

The reviewer is correct that the numbers do not seem to match. However, the text describes all patients with disease recurrence during the trial (including the follow-up phase up to 5 years). The consort diagram ends after completion of the 3rd cycle of injections (week 61; treatment phase). It was chosen not to include end of study (5-year follow-up) in the consort diagram as not all patients completed the 5 years yet so numbers are subject to change with longer follow-up.

5) Remove percentages next to reporting the total who died. Using raw percentages ignores censoring and is misleading.

We agree with the reviewer that the raw percentages of patients who died could be considered misleading as many patients are censored before the complete 5-year follow-up, and the percentages are therefore expected to rise with longer follow-up. As the same holds true for recurrence of disease (to a lower extent due to less censoring of data), we removed this information from the manuscript.

6) In the "Adverse events" section it lists that 5% and 6% of study-related grade 3 adverse events were reported in the nDC and control groups, respectively. Where are these percentages shown in Table 2? They do not align.

The reviewer rightly states that the study-related adverse events as mentioned in the text are not presented in Table 2. In the table the study-related adverse events were divided in apheresis-related and treatment-related adverse events. The combined parameter is now included in Table 2 and explained in the legend. As one patient in the nDC treatment group had both a grade 3 adverse event related to apheresis and one related to treatment, the total study-related percentage is lower than the 2 combined.

7) In the Discussion it states that "The main conclusion of this trial, the lack of improvement in the 2-year RFS rate with nDC therapy, was not impacted by the incomplete accrual of patients". How can the authors be sure of this when the accrual was stopped? This lower sample size was not powered to detect this conclusion. This should be reworded.

What we meant to say was the following: accrual of the trial was stopped prematurely, reaching accrual of 148 evaluable patients of the planned 210 patients (70% accrual). As the current RFS data show a HR of 1.25 favoring placebo it is very unlikely that full accrual would have turned the HR in favor of nDC therapy with a significant difference between the arms. Therefore, we concluded that the incomplete accrual did not impact the outcome of the trial. However, we agree that the quoted statement could be misleading, and we have therefore reworded this sentence in the discussion for better clarity.

8) A limitation should be added regarding that the sample size powered in the protocol was not achieved, thus this may affect the conclusions of the trial.

As suggested by the reviewer we rephrased the sentence regarding the impact of the incomplete accrual of the trial on the outcome.

9) Why were patients with an unknown primary tumor also eligible for inclusion? Does this make the population heterogenous? Please elaborate in the "Study design and participants" section.

Unknown primary tumors represent 2-9% of all cases with advanced melanoma and may be partially explained by a regressed primary cutaneous melanoma. Nonetheless, they

are regarded as cutaneous melanoma. Literature on melanoma patients with unknown primary tumors shows a quite similar prognosis to patients with a known primary cutaneous melanoma (reference: Bae et al, JAAD 2015, doi.org/10.1016/j.jaad.2014.09.029). This is in stark contrast to patients with a primary mucosal or uveal melanoma which have a different prognosis and respond differently to systemic treatment. Therefore, it is common to include patients with an unknown primary tumor in cutaneous melanoma studies but exclude uveal and mucosal melanomas, which is in line with our protocol.

10) The protocol specifies reporting a 90% confidence interval, but the manuscript "Statistical analysis" section reports using 95% confidence intervals. This should be harmonized.

We thank the reviewer for pointing out the inconsistency between the protocol and the manuscript. The protocol indeed states a 90% confidence interval for the median survival and hazard ratios. We have adjusted the confidence intervals throughout the manuscript. These adjustments have no impact on the primary outcome of the trial and no statistical differences are present with these less strict intervals.

11) What is meant by "P values of point estimates" in the "Statistical analysis" section? Please clarify and elaborate.

We meant "p-values for differences between fractions", which was indeed not clear from this wording. We have explained this in more detail in the revised manuscript.

12) Survival was censored at last follow up or time of switch to adjuvant therapy for those in the control arm that switched. Please rectify this in the "Statistical analysis" section.

Thanks for pointing this out, we have reworded this as suggested.

Reviewer #2 (Remarks to the Author): with expertise in melanoma, immunotherapy, clinical trials

The investigators have performed a phase III placebo-controlled clinical trial in 148 patients with resected stage IIIB/C melanoma, with 2:1 randomization to treatment with natural DCs vs. placebo. Patients were stratified by stage, radiotherapy, BRAF status, and HLA-A0201 status. The nDCs were loaded with tumor antigens and injected intranodally. Adverse events were similar for treatment and placebo. T cell responses were detected in most patients, but there was no significant difference in recurrence-free survival. This is a new approach using natural circulating DCs (nDCs) rather than monocyte-derived DCs generated in vitro. They show that the DCs primarily include conventional DC2s (cDC2s) and plasmacytoid DCs (PDCs). A prior feasibility study by this group supported the study design. Patients with AJCC v7 stage IIIB-C melanoma were enrolled. The primary endpoint was 2 year RFS. Enrollment was discontinued early because of new approved therapies. This paper reports the final results with of this trial. The nDC therapy was feasible, with only about 4% not treated due to unsuccessful apheresis or recurrence before the first injection. The groups were well-matched. AEs were similar in the DC and placebo groups, supporting safety of this vaccine strategy.

The nDC were loaded with overlapping peptide pools of MAGE-A3 and NY-ESO-1 during overnight culture with GM-CSF, IL-3 and KLH, then with 14 peptides from melanocytic differentiation antigens (gp100 and tyrosinase) and cancer testis antigens (MAGE-C2, MAGE-A3, NY-ESO-1), including 11 short peptides restricted by Class I MHC (6 HLA-A2.1), and 3 longer peptides presented by class II MHC.

The RFS was no different between nDC and placebo, with a weak trend to shorter RFS with the DC vaccine. Similarly, OS was not different, with an even weaker trend favoring placebo.

T cell responses to vaccine antigens were assessed by measuring IFN γ production of lymphocytes cultures from a skin test after the first cycle. Of those evaluable by this method, there was a dramatically higher T cell response rate in the DC vaccine group. Overall, 67% of the 90% evaluable by this method had T cell responses in this assay.

The authors are to be congratulated on performing what appears to be the first randomized multicenter placebo controlled trial of nDC + peptides vs placebo. The study was closed with about 70% of target enrollment, but the lack of clinical benefit for the nDC arm is not likely to be have been altered by completing the target enrollment. The presented data certainly support immunogenicity of the nDC vaccine.

There are some aspects of the study that might be enhanced to understand the outcomes more clearly.

1) The immune analysis is based on in vitro culture of T cells accumulating at a site of skin test (DTH) and stimulated with antigen-pulsed PBMC. While these data support immunogenicity in the nDC patients, compared to the placebo patients, it would be helpful to use a measure of immunogenicity that may be compared to prior experience with other vaccine strategies, such as analysis ex vivo of circulating T cells. Was blood collected that could be assayed by ELISpot or other functional assays?

We have previously shown that the DTH skin test is more likely to detect antigen-specific immune responses and besides cytokine production also proofs the migratory capacity of the T cells towards tissue (references: de Vries et al, JCO 2015, DOI: 10.1200/JCO.2005.06.478; Aarntzen et al, Cancer Res 2012, doi.org/10.1158/0008-5472.CAN-12-2479). Ultimately, migration in peripheral tissue is essential for an antitumor effect and thus the DTH test results might be more predictive for functional responses than assays on blood cells. For comparison with previous DC-based immunotherapy trials, we have included the percentage of functional tumor antigen-specific T cells in the introduction and discussion. The immunological response rate, with functional tumor-specific T cells detected in DTH skin tests in two-thirds (67%) of the patients with nDC therapy in this study, are in line with our previous results in stage III melanoma patients (64%).

For comparison with other vaccination strategies, as requested by the reviewer, we included the results from the MHC dextramer stainings of both the skin-test infiltrating CD8 T cells and peripheral blood CD8 T cells of HLA-A1, HLA-A2 and HLA-B35 positive patients.

Strikingly, baseline presence of antigen-specific T cells in the blood was present in 43% of patients in the control group compared to only 14% of patients in the treatment group.

2) Also, the T cell responses are not detailed in terms of antigen reactivity or CD8 vs CD4 T cell reactivities. Since most patients were HLA-A2.1 negative, and since 15mer peptides may not induce CD8 T cells as readily as CD4 T cells, it would be helpful to understand the immune reactivity as a function of HLA expression.

The reviewer is correct that the assay to detect IFN γ production is detecting CD8 and CD4 T cell responses. For detecting antigen-specific CD8 T cells, we performed MHC class I dextramer stainings and included the results in the manuscript. In SKILs, 17 out of 48 patients in the nDC treatment group showed dextramer positive CD8 T cells. We more frequently detected IFN γ production compared to dextramer positivity in the SKILs (67% vs 35% in the nDC treatment group). Based on these data, we conclude that both CD8 as well as CD4 T cell responses were induced by the nDC.

3) The nDC used in this approach were primarily cDC2 and PDC. These are certainly good to include, but cDC1 are commonly considered best for induction of CD8 T cells either directly or after cross-presentation. What were the reasons for not including cDC1s?

We agree with the reviewer that cDC1 are an interesting subset for further research as they are considered the most potent stimulators of cytotoxic T cells as specialized cells in cross presentation. GMP isolation of cDC1s was not available when this trial was designed/started and therefore these cells were not included in the trial. Together with Miltenyi Biotec we developed the procedure for clinical-grade isolation of these cells and recently the first clinical trial with intratumor injection of this subset were published (Schwarze et al, JITC 2022, DOI: 10.1136/jitc-2022-005141).

4) Since the presumption of the authors that nDC are preferable to use, compared to monocyte-derived DC, do the authors have data on markers of DC activation and maturation, and cytokine production compared to that achieved with monocyte-derived DC? Also the potency of the mature DC appears low in panel D of Supplemental Figure 3, panel D: is there a comparator group?

The use of nDCs in cell-based immunotherapy is more practical compared to moDC due to the shorter culture time and automated closed GMP-grade manufacturing. This is crucial for a treatment to be suitable for standard of care treatment and application in other centers.

We have no data available of nDC and moDC markers cultured from the same patients. Compared to another cohort, the potency index of the nDC patients in this study seems similar, possibly better, than monocyte-derived DC (see figure below). As this is a comparison with another cohort of patients (unpublished data) this will not be incorporated in the manuscript.

As for the maturation markers of nDC vs moDC, we compared with another cohort of cancer patients which received adjuvant moDC treatment. As shown below, the

maturation markers CD80, CD83, and CD86 are more highly expressed on moDC (see figure below; markers indicated with * are on moDC). However, the nDC have a huge advantage as they produce type I IFN and IL-12, where moDC matured with cytokines and PGE2 do not (Boullart et al, Cancer Imm Imm 2008, DOI: 10.1007/s00262-008-0489-2).

5) The discussion addresses some relevant considerations, but it would be valuable to elaborate on whether lack of clinical benefit may relate to the magnitude or function of the induced T cells. The companion paper about microbiome is interesting, but it is difficult to understand why there would be a significant mismatch in microbiome in a multicenter trial.

We thank the reviewer for additional suggestions for possible explanations of the lack of clinical translation of the immunological results. We have added this hypothesis to the discussion.

On the second question, we agree with the reviewer that one would not expect a major mismatch in baseline characteristics (including the microbiome) in a randomized trial but by chance, and in theory an infinite amount of possible baseline characteristics, it is not uncommon. To avoid an imbalance in baseline characteristics by chance, trials stratify for the most important *known* prognostic of predictive factors. In this trial patients were therefore stratified according to stage of the disease, adjuvant radiotherapy, BRAF mutation status, and HLA-type. In addition to the difference in microbiome at baseline, as described in the companion paper, we also see a difference in the number of patients with tumor antigen-specific T cells in the blood at baseline (13.6 vs 42.8%) which could have further impacted the outcome of the trial.

Also, a few minor comments:

a) The Kaplan-Meier curve in Figure 2A is labelled Progression-free survival, but probably should be corrected to "recurrence-free survival".

The reviewer is correct. The title on the y-axis of the KM curve in figure 2A is adjusted to recurrence-free survival.

b) Please clarify details about the overlapping peptide pools for NY-ESO-1 and MAGE-A3? Were these 15 mers overlapping by 11?S

The reviewer is correct. The NY-ESO-1 and MAGE-A3 peptivators are a pool of overlapping peptides, consisting mainly of 15-mer sequences with 11 amino acids overlap, covering the complete sequence of the protein. We have added this additional information to the methods.

c) The data in S Fig 3, Panel A are not clear: It is not clear what the Y axis represents in panel A.

The purity of the nDC product is defined as the percentage of nDCs (sum of CD123+BDCA2+ pDC and cD1c+CD20- cDC2) of all alive cells. This is now incorporated in the methods. Minor impurities are usually present in the nDC product and consist of small amounts of lymphocytes and monocytes.

d) It would be helpful to add a protocol schema with a timeline of interventions, to follow what was done in the study.

The study timeline is depicted in Supplementary Figure 1. For reader convenience the figure is now added to the methods section (Figure 4).

Reviewer #3 (Remarks to the Author): with expertise in melanoma, immunotherapy, clinical trials

This is an unsuccessful phase III clinical trial of a DC product. The authors claim that this is being presented such that a companion paper describing the translational science of why the trial was unsuccessful is placed into the proper context.

1. Why was apheresis unsuccessful in two patients?

In both patients we were not able to establish peripheral access for apheresis despite multiple attempts and ultrasound guidance. According to protocol, it was not allowed to place a central line for apheresis. We added this information to the results section.

2. Table 1: Unfortunately, the investigators still used AJCC version 7. Lymph node disease suggests better patient characteristics in the treated arm. It would be worthwhile to comment on outcome based on lymph node status in the discussion.

The protocol was written prior to the 8th edition of the AJCC staging system. As most adjuvant trials in melanoma are performed with 7th edition it might have even been beneficial over the 8th edition for comparison (which is not relevant as the trial did not meet its primary endpoint). The baseline characteristics, including lymph node status, are similar between the groups (p-value added to Table 1). We believe the study is too small for meaningful subgroup analyses and it would not change the primary outcome of the trial.

3. It is noted that the study failed to meet the endpoint and there may be a slight disadvantage in the DC-treated group. There was no clinical benefit of IFN γ producing antigen specific T cells. The authors speculated that this is secondary to gut microbiome, but there should be more of a link to the companion paper. The clinical trial paper should be concise as a stand-alone paper.

We aim to publish the gut microbiome data as a companion paper in the same issue. To be more concise as a stand-alone paper we have added the time of gut microbiome collection in the methods and elaborated a bit more on the data of the companion paper in the discussion.

4. What was the grade 4 reaction in the treatment group?

The patient with a grade 4 adverse event was admitted to another hospital with an acute coronary syndrome with pre-syncope suggestive of hemodynamic instability. The adverse event occurred over 15 months after the last nDC injection and was considered unlikely related to the study treatment. As it concerns an unrelated adverse events we did not add the above information to the manuscript.

5. The authors noted: "Significantly more functional antigen-specific T cells were detected in the nDC treatment group, indicative that the nDCs were able to induce immune response directed against the tumor-associated antigens presented". This should be tested via an assay.

The presence of functional antigen-specific T cells was tested by IFN γ production in the DTH skin test as described in the manuscript. The statistical difference in the presence of functional antigen-specific T cells in the treatment groups (67.1% vs 3.8%) was tested with a chi-square test ($p < 0.001$).

6. Supplementary figure 3 should also be divided into the two groups

The control group did receive placebo injections consisting of NaCl supplemented with Albuman. As the control group did not receive unloaded/unmatured nDCs, no nDC product was manufactured for these patients. Therefore, supplementary figure 3 contains nDC product information of the nDC treatment group only.

7. Minor comment: Personal preference would be that supplementary Figure 1 is in the manuscript. The description of the trial is essential to have readily available to the reader of the paper. It could be combined with the current Figure 1.

As 2 reviewers requested the treatment schedule presented in supplementary figure 1 to be more prominent, we decided to add the figure in the method section of the manuscript (Figure 4). If the editor would like to have the figure elsewhere in the manuscript, we are willing to move the figure to a more appropriate place.

REVIEWERS' COMMENTS

Reviewer #1 (Remarks to the Author):

Thank you to the authors for addressing the comments.

Regarding comment 2, I think the authors should mention that there was 3 patients received less than 6 doses of nDC injections regardless of whether or not this impacts the efficacy. I think will helpful for others to know how likely it may be that low nDC yield can occur. Otherwise all other comments have been adequately addressed.

Reviewer #2 (Remarks to the Author):

The investigators have performed a phase III placebo-controlled clinical trial in 148 patients with resected stage IIIB/C melanoma, with 2:1 randomization to treatment with natural DCs vs. placebo. Patients were stratified by stage, radiotherapy, BRAF status, and HLA-A0201 status. The nDCs were loaded with tumor antigens and injected intranodally. Adverse events were similar for treatment and placebo. T cell responses were detected in most patients, but there was no significant difference in recurrence-free survival. This is a new approach using natural circulating DCs (nDCs) rather than monocyte-derived DCs generated in vitro. They show that the DCs primarily include conventional DC2s (cDC2s) and plasmacytoid DCs (PDCs). A prior feasibility study by this group supported the study design. Patients with AJCC v7 stage IIIB-C melanoma were enrolled. The primary endpoint was 2 year RFS. Enrollment was discontinued early because of new approved therapies. This paper reports the final results with of this trial. The nDC therapy was feasible, with only about 4% not treated due to unsuccessful apheresis or recurrence before the first injection. The groups were well-matched. AEs were similar in the DC and placebo groups, supporting safety of this vaccine strategy.

The nDC were loaded with overlapping peptide pools of MAGE-A3 and NY-ESO-1 during overnight culture with GM-CSF, IL-3 and KLH, then with 14 peptides from melanocytic differentiation antigens (gp100 and tyrosinase) and cancer testis antigens (MAGE-C2, MAGE-A3, NY-ESO-1), including 11 short peptides restricted by Class I MHC (6 HLA-A2.1),

and 3 longer peptides presented by class II MHC.

The RFS was no different between nDC and placebo, with a weak trend to shorter RFS with the DC vaccine. Similarly, OS was not different, with an even weaker trend favoring placebo. T cell responses to vaccine antigens were assessed by measuring IFN γ production of lymphocytes cultures from a skin test after the first cycle. Of those evaluable by this method, there was a dramatically higher T cell response rate in the DC vaccine group. Overall, 67% of the 90% evaluable by this method had T cell responses in this assay. The authors are to be congratulated on performing what appears to be the first randomized multicenter placebo controlled trial of nDC + peptides vs placebo. The study was closed with about 70% of target enrollment, but the lack of clinical benefit for the nDC arm is not likely to have been altered by completing the target enrollment. The presented data support safety and immunogenicity of the nDC vaccine, but not clinical efficacy.

The authors have replied to questions about the immune analyses, and those replies are helpful and appreciated; however, they raise other questions. Analyses of peripheral blood for T cell responses were not done routinely, The authors offer prior data that thee SKILs are a more sensitive measure of T cell response and that they represent T cells that have migrated to skin and thus may be more meaningful than circulating T cells. The authors have added new data on dextramer staining for some Class I MHC-restricted T cells, and interestingly detect a trend to more at baseline in the control patients than in the DC arm (not significant, $p = 0.13$). Overall, it does appear that the nDC vaccines were immunogenic, but the immune response assessment would be stronger if it had data on T cell responses in blood, to correlate with the SKIL data, and it would be valuable to know understand CD4 and CD8 T cell responses. The dextramer data are stated in a paragraph, but the actual data are not evident. It would be valuable to include supplemental data with the dextramer flow plots to understand what is considered positive, and also to have data broken down by patient, for example to understand to what extent pre-existing T cell responses by dextramer were in the same or different patients with SKIL responses after vaccination.

The authors' replies about DC markers, and about focusing on cDC2 and PDC are helpful, but it would be valuable to add to the manuscript some of the text in the response to the reviews, to explain that recent data on DC with cDC1 might affect future considerations.

Reviewer #3 (Remarks to the Author):

Agree not to include p values inside table.

Response to the reviewers MIND-DC

Reviewer #1 (Remarks to the Author):

Thank you to the authors for addressing the comments.

Regarding comment 2, I think the authors should mention that there was 3 patients received less than 6 doses of nDC injections regardless of whether or not this impacts the efficacy. I think will helpful for others to know how likely it may be that low nDC yield can occur. Otherwise all other comments have been adequately addressed.

Response:

The suggestion of the reviewer was already addressed in the revised version of the manuscript, by stating the following in the paragraph on nDC product characteristics: "All patients receiving a placebo injection in the nDC treatment group received at least nDC injections, and 20 out of 23 patients received at least 6 nDC injections (range of nDC injections: 3-8)."

Reviewer #2 (Remarks to the Author):

The investigators have performed a phase III placebo-controlled clinical trial in 148 patients with resected stage IIIB/C melanoma, with 2:1 randomization to treatment with natural DCs vs. placebo. Patients were stratified by stage, radiotherapy, BRAF status, and HLA-A0201 status. The nDCs were loaded with tumor antigens and injected intranodally. Adverse events were similar for treatment and placebo. T cell responses were detected in most patients, but there was no significant difference in recurrence-free survival. This is a new approach using natural circulating DCs (nDCs) rather than monocyte-derived DCs generated in vitro. They show that the DCs primarily include conventional DC2s (cDC2s) and plasmacytoid DCs (PDCs). A prior feasibility study by this group supported the study design. Patients with AJCC v7 stage IIIB-C melanoma were enrolled. The primary endpoint was 2 year RFS. Enrollment was discontinued early because of new approved therapies. This paper reports the final results with of this trial. The nDC therapy was feasible, with only about 4% not treated due to unsuccessful apheresis or recurrence before the first injection. The groups were well-matched. AEs were similar in the DC and placebo groups, supporting safety of this vaccine strategy.

The nDC were loaded with overlapping peptide pools of MAGE-A3 and NY-ESO-1 during overnight culture with GM-CSF, IL-3 and KLH, then with 14 peptides from melanocytic differentiation antigens (gp100 and tyrosinase) and cancer testis antigens (MAGE-C2, MAGE-A3, NY-ESO-1), including 11 short peptides restricted by Class I MHC (6 HLA-A2.1), and 3 longer peptides presented by class II MHC.

The RFS was no different between nDC and placebo, with a weak trend to shorter RFS with the DC vaccine. Similarly, OS was not different, with an even weaker trend favoring placebo.

T cell responses to vaccine antigens were assessed by measuring IFN γ production of lymphocytes cultures from a skin test after the first cycle. Of those evaluable by this method, there was a dramatically higher T cell response rate in the DC vaccine group. Overall, 67% of the 90% evaluable by this method had T cell responses in this assay.

The authors are to be congratulated on performing what appears to be the first randomized multicenter placebo controlled trial of nDC + peptides vs placebo. The study was closed with about 70% of target enrollment, but the lack of clinical benefit for the nDC arm is not likely to have been

altered by completing the target enrollment. The presented data support safety and immunogenicity of the nDC vaccine, but not clinical efficacy.

The authors have replied to questions about the immune analyses, and those replies are helpful and appreciated; however, they raise other questions. Analyses of peripheral blood for T cell responses were not done routinely, The authors offer prior data that thee SKILs are a more sensitive measure of T cell response and that they represent T cells that have migrated to skin and thus may be more meaningful than circulating T cells. The authors have added new data on dextramer staining for some Class I MHC-restricted T cells, and interestingly detect a trend to more at baseline in the control patients than in the DC arm (not significant, $p = 0.13$). Overall, it does appear that the nDC vaccines were immunogenic, but the immune response assessment would be stronger if it had data on T cell responses in blood, to correlate with the SKIL data, and it would be valuable to know understand CD4 and CD8 T cell responses. The dextramer data are stated in a paragraph, but the actual data are not evident. It would be valuable to include supplemental data with the dextramer flow plots to understand what is considered positive, and also to have data broken down by patient, for example to understand to what extent pre-existing T cell responses by dextramer were in the same or different patients with SKIL responses after vaccination.

The authors' replies about DC markers, and about focusing on cDC2 and PDC are helpful, but it would be valuable to add to the manuscript some of the text in the response to the reviews, to explain that recent data on DC with cDC1 might affect future considerations.

Response:

As suggested by the reviewer we included a supplementary table in which we show T cell specificity in blood and DTH per patient and per antigen (Supplementary Figure 4). We noticed minor inconsistencies in the immunological data we added during the revision which we corrected to adhere to our prespecified endpoint based on the immune population (patient who underwent a DTH skin test only). Furthermore, we added examples of dextramer flow plots per epitope to give insight in what is considered positive (Supplementary Figure 3). Lastly, we included the role of the DC subsets in the discussion, introducing the recent insight on cDC1 treatment.

Reviewer #3 (Remarks to the Author):

Agree not to include p values inside table.

Response:

No further requests from the reviewer.